# Three-Dimensional-Bioprinted Embedded-Based Cerebral Organoids: An Alternative Approach for Mini-Brain In Vitro Modeling Beyond Conventional Generation Methods

**DOI:** 10.3390/gels11040284

**Published:** 2025-04-11

**Authors:** Rosalba Monica Ferraro, Paola Serena Ginestra, Miriam Seiti, Mattia Bugatti, Gabriele Benini, Luana Ottelli, William Vermi, Pietro Luigi Poliani, Elisabetta Ceretti, Silvia Giliani

**Affiliations:** 1“Angelo Nocivelli” Institute for Molecular Medicine, Department of Molecular and Translational Medicine, University of Brescia, Viale Europa 11, 25123 Brescia, Italy; gabriele.benini@unibs.it (G.B.); luana.ottelli.bs@gmail.com (L.O.); silvia.giliani@unibs.it (S.G.); 2ASST Spedali Civili, Piazzale Spedali Civili 1, 25123 Brescia, Italy; mattiabugatti16@gmail.com (M.B.); william.vermi@unibs.it (W.V.); luigi.poliani@unibs.it (P.L.P.); 3Department of Mechanical and Industrial Engineering, University of Brescia, Via Branze 38, 25123 Brescia, Italy; paola.ginestra@unibs.it (P.S.G.); miriam.seiti@unibs.it (M.S.); elisabetta.ceretti@unibs.it (E.C.); 4Anatomy and Pathological Histology Section, Department of Molecular and Translational Medicine, University of Brescia, Viale Europa 11, 25123 Brescia, Italy; 5National Center for Gene Therapy and Drugs Based on RNA Technology-CN3, 25123 Brescia, Italy

**Keywords:** cerebral organoids, induced pluripotent stem cells, neural stem cells, hydrogel, 3D bioprinting, tissue engineering

## Abstract

Cerebral organoids (cORGs) obtained from induced pluripotent stem cells (iPSCs) have become significant instruments for investigating human neurophysiology, with the possibility of simulating diseases and enhancing drug discovery. The current approaches require a strict process of manual inclusion in animal-derived matrix Matrigel^®^ and are challenged by unpredictability, operators’ skill and expertise, elevated costs, and restricted scalability, impeding their extensive applicability and translational potential. In this study, we present a novel method to generate brain organoids that address these limitations. Our approach does not require a manual, operator-dependent embedding. Instead, it employs a chemically defined hydrogel in which the Matrigel^®^ is diluted in a solution enriched with sodium alginate (SA) and sodium carboxymethylcellulose (CMC) and used as a bioink to print neural embryoid bodies (nEBs). Immunohistochemical, immunofluorescence, and gene expression analyses confirmed that SA-CMC-Matrigel^®^ hydrogel can sustain the generation of iPSC-derived cortical cORGs as the conventional Matrigel^®^-based approach does. By day 40 of differentiation, hydrogel-based 3D-bioprinted cORGs showed heterogeneous and consistent masses, with a cytoarchitecture resembling an early-stage developmental fetal brain composed of neural progenitor cells PAX6^+^/Ki67^+^ organized into tubular structures, and densely packed cell somas with extensive neurites SYP^+^, suggestive of cortical tissue-like neuronal layer formation.

## 1. Introduction

In recent years, the cross-interaction between biomedical research and tissue engineering has led to three-dimensional (3D) in vitro modeling, and its generation has revolutionized the way we understand and interact with complex biological systems. The push in this direction aims to better mimic human morphology and physiology at the multiscale from the molecular to the organ level, passing through cellular and tissue levels. The ability to create and manipulate 3D cellular models of both pluripotent and differentiated cells has significantly advanced research, development, and clinical applications due to crucial biological features absent in 2D biological systems. Traditional monolayer cell cultures, in which cells are grown in a flat, 2D plane, have indeed long been a staple in cell biology. However, they fail to mimic the complex 3D architecture which cells experience in their native environment, where cell–cell and cell–matrix interactions are fundamental requirements [1]. The microenvironment in pluripotent stem cell lines is of particular importance. These cells, such as induced pluripotent stem cells (iPSCs) or embryonic stem cells (ESCs), demonstrate the abilities to self-renew and to self-organize under various conditions to form complex 3D tissue structures called organoids, with advanced morphological and functional fidelity to the in vivo counterpart. Inducing stem cells to differentiate into 3D organoids involves a series of well-defined conditions that mimic the in vivo developmental environment. These conditions vary depending on the type of organoid being generated (e.g., brain, liver, gut, or kidney) and usually involve specific growth factors or cytokines, particular coating surfaces, and mechanical and physical factors such as oxygen levels, shear stress, temperature, and pH. Specifically, an organoid is a 3D structure that spontaneously self-organizes into adequately differentiated functional cell types and can recapitulate the functions of the target organ [2]. Since the development of the first intestinal organoids in 2009, research has increasingly expanded the range of tissues and organs that can be modeled, starting from iPSCs, like the liver [3], heart [4], mammary gland [5], pancreas [6], and brain [7]. Especially for tissues with difficulties in having a biopsy, such as the nervous tissue, there is a particular need for the development of iPSC-derived in vitro disease model that can fill the gap and lead the study of neurological disorders that can present with a variety of cognitive, social, and motor impairments, and multifactorial genetic disorders. The stem cell niche, a sensitive milieu made up of stem cells as well as supporting stromal cells, extracellular matrix, and growth factors, is not well represented by 2D cultures, as was previously noted. Therefore, the iPSCs’ potential combined with the modern 3D culturing technologies may enable to generation and exploitation of human brain-like tissues, named cerebral organoids or mini-brains. Specifically, a human cerebral organoid is described as a self-assembled sphere with an internal cytoarchitecture reminiscent of a laminated human neocortex and transcriptionally equivalent to a mid-fetal prenatal human brain [8]. Mini-brains can be derived from patient-specific iPSCs and recapitulate key aspects of human brain development. Therefore, they have become invaluable tools for studying neurodevelopment, disease modeling, and drug discovery. Over the years, various protocols have been developed to generate cerebral organoids from iPSCs [9], each with distinct methods for mimicking developmental processes occurring in vivo. A widely recognized source of cerebral organoid protocols is from Lancaster’s group, which has pioneered some of the most influential techniques. Their protocols aim to closely replicate the neurodevelopmental stages of the human brain, from neural induction and patterning to cortical development and regionalization [10]. These workflows typically involve the induction of iPSC-derived neuroepithelial structures using defined growth factors. These structures are then embedded in a 3D extracellular environment, often matrix Matrigel^®^, to facilitate self-organization and guide iPSC differentiation through the key stages of neural development. Matrigel^®^ is an expensive xenobiotic extracellular matrix (ECM) derived from Engelbreth-Holm-Swarm mouse sarcomas, used to mimic the basement membrane. It serves as a supportive scaffold, helping organoids maintain their 3D structure while facilitating nutrient and growth factor diffusion, thus mimicking the in vivo microenvironment. This embedding not only preserves the organoids’ cellular architecture but also promotes cellular interactions that are essential for proper brain development, including neurogenesis, synaptogenesis, and the formation of functional neuronal networks [11]. Although the matrix Matrigel^®^ can promote cell growth and efficient differentiation, it exhibits heterogeneity, batch-to-batch variability, and a poorly defined protein composition that can determine the uncontrollability of cellular niches and decrease the reproducibility of organoids. Despite its versatility, Matrigel^®^ is a hugely complex matrix, with an estimated number of unique proteins more than 1800 [12]. This variety of components, combined with random batch-to-batch differences, makes it difficult to identify specific signals for organoid development and function. Furthermore, a high quantity does not necessarily equate to suitability. Although the Matrigel^®^ matrix contains a wide range of proteins, it does not always provide all the essential substrates required for proper organoid formation. For instance, gut organoids cultured in Matrigel^®^ do not show the typical morphology of mammalian intestines due to the sub-optimal amount of laminin 511 [13]. Moreover, Matrigel^®^ does not fully mimic the complex and dynamic nature of the human brain’s extracellular environment, which can limit the physiological relevance of brain organoids in disease modeling [14]. For example, the absence or low levels of human-specific ECM proteins, such as tenascin-C, fibronectin, and laminin isoforms, particularly laminin-511 and laminin-521, and the lack of human-specific proteoglycans like brevican, neurocan, and versican, and heparan sulfate proteoglycans. Moreover, Matrigel shows batch-to-batch variability in stiffness, which affects reproducibility in brain organoid cultures. It also lacks tunability (elastic modulus (E) typically ranges from 100 to 400 Pa), which is softer than the developing human brain ECM (~500–1000 Pa), potentially limiting the proper mechano-transduction cues for neural differentiation. Therefore, identifying and developing alternative materials, such as synthetic bioinks [15], decellularized matrices [16], or bioengineered scaffolds [17,18,19], is essential to overcome these limitations. The limitations of 2D cultures, including their restricted cell adherence to the extracellular matrix (ECM), non-physiological surfaces (such as glass or plastic), and irregular cell-to-cell contacts, have been addressed by the development of these final systems. By precisely arranging cells to resemble the structure of actual tissues, bioprinting effectively connects in vitro illness models with in vivo pre-clinical models [20]. Bioprinting is an additive manufacturing process that produces 3D models in the macro- (~1-10-100 mm) and microscale (~100 μm), through the deposition of several layers of bioinks and living cells to obtain tissue-like structures [21,22,23,24]. Different polymers are available to produce a bioink; most of them are from natural sources (i.e., gelatin, alginate, collagen, cellulose, etc.), but valid synthetic alternatives have been demonstrated (i.e., PEG, PVA, and PVP) [25]. The type of tissue or cell lineage that is of interest for its mechanical and chemical qualities determines which material is used. Research has shown that varying substrate stiffness results in altered morphologies and cell adhesion [26,27,28]. The bioink used in extrusion-based bioprinting is forced out through a nozzle or needle and applied to the chosen substrate, which cannot be self-supporting, particularly if it is made of soft materials. To facilitate bonding between polymer chains and produce a compact structure, a crosslinking step is often carried out [29,30]. Therefore, consistent substrates with adjustable stiffness depending on the targeted cellular line may be obtained from such a technique. These substrates usually contain encapsulated cells (cell-laden bioinks), which can be arranged in a 3D spatial configuration resembling the intrinsic 3D targeted tissue organization [31,32]. Although the approach used by Lancaster [33] to generate cerebral organoids was highly effective, it presents challenges due to its complexity, the need for expensive neural induction components, and the use of xenobiotic extracellular matrix materials. Lancaster’s protocol for embedding neural embryoid bodies (nEBs) in Matrigel^®^ is a complex manual procedure, with its success highly dependent on the technician’s technical skills and expertise.

In this study, we present a novel method for generating brain organoids that overcomes these limitations. Our approach does not require a manual, operator-dependent embedding. Instead, it employs a chemically defined hydrogel in which the Matrigel^®^ is diluted (1:10) in a solution enriched with 1.5 *w*/*v*% sodium alginate (SA), 1 *w*/*v*% of sodium carboxymethylcellulose (CMC) in Milli-Q water and used as a bioink to print nEBs. SA and CMC were chosen to compose the hydrogels because these components offer a more controllable and biomimetic alternative to Matrigel^®^ for brain organoid culture. SA hydrogels can be precisely adjusted in stiffness through ionic crosslinking, better matching the elasticity of human brain ECM. CMC enhances viscosity and printability, ensuring structural integrity during fabrication [34,35]. Beyond mechanics, SA-CMC hydrogels can be potentially chemically functionalized with human-relevant ECM proteins. This customization supports stem cell adhesion, differentiation, and network formation while providing a stable and reproducible environment for long-term studies [36]. Immunohistochemical, immunofluorescence, and gene expression analyses confirmed the generation of brain organoids with features of the forebrain, midbrain, and hindbrain regions. This simplified system offers significant potential for both basic research and translational applications, where process standardization is essential for generating reproducible in vitro models of neurological diseases.

## 2. Results and Discussion

### 2.1. iPSC-Derived Cerebral Organoid Generation by Manual Matrigel^®^ Embedding 

The development of 3D cellular models, particularly cerebral organoids (cORGs) derived from iPSCs, has significantly advanced biomedical research by providing a more physiologically relevant representation of human brain development and pathology than traditional 2D cell cultures. cORGs replicate key aspects of neurodevelopment, making them invaluable tools for studying neurological disorders and testing potential personalized therapeutic strategies that would otherwise be limited to post-mortem samples. We set up a 40-day protocol for differentiating iPSCs into cORGs, structured into four key steps: iPSC-derived neural embryoid body (nEB) generation (5 days), neuroectodermal induction (2 days), matrix embedding for neuroepithelium expansion (3 days), and cerebral organoid maturation (30 days) (Figure 1A). Specifically, cORGs were generated by optimizing the protocol described by Lancaster [10,31] using commercial media enriched with growth factors and supplements. iPSC colonies were dissociated into single cells and cultured into AggreWell™ Microwell Culture Plates, enabling the scalable production of nEBs with uniform shape and size, each consisting of approximately 6000 cells. We daily monitored the growth and development of cORGs, checking the changes in size, shape, and color (Figure 1B). On day 1, the iPSC-derived nEBs appeared spherical with a dense core and loose cells at the edge. Chemical induction promoted the thickening of the peripheral region of the spheroids, which by day 7 showed a well-defined compact edge and an approximate diameter of 200 μm. On day 7, the manual Matrigel^®^ embedding of nEBs was performed, a procedure that is both technically demanding and requires strict sterility maintenance. It requires the use of a wide-bore 200 µL pipette tip (obtained by cutting a tip with a sterile scalpel) to quickly transfer each micrometric organoid onto a sheet of dimpled parafilm (sterilized with ethanol). After removing the excess medium, each dry organoid is covered with a small volume of Matrigel^®^, using a pipette with a cold tip, to prevent premature gelation. Finally, after 30 min in an incubator at 37 °C, using sterile forceps, the embedding surface containing the Matrigel^®^ droplets is lifted and moved directly above a well of a 6-well ultra-low adherent plate. The Matrigel^®^ droplets are then gently washed off the sheet into the well using the medium. Around day 10, embedded nEBs developed expanded neuroepithelia as evidenced by the budding of the organoid surface. These buds, known as neural rosettes, resemble the folding structure of the cerebral ventricles. As reported in Figure 1B, from day 20, a progressive increase in organoid size was observed as the structure expanded to colonize the available matrix area, becoming increasingly dense and darker under an optical microscope.

### 2.2. Manually Matrigel^®^-Embedded Cerebral Organoid Characterization

After 4 weeks of maturation, the manually Matrigel^®^-embedded cORGs were analyzed in terms of neuro-marker expression by immunohistochemistry (IHC) and quantitative PCR (qPCR). To evaluate the neuronal induction efficiency, maturation degree, regionalization, and cellular heterogeneity of the cORGs obtained, the main identifying markers of macro-areas of the brain in development were selected. IHC confirmed not only the expression of the expected markers, but also their localization and compartmentalization (Figure 2A). Hematoxylin/Eosin staining (HE) revealed two different regions: a parenchyma area with lower cell density and scattered nuclei immersed in the extracellular matrix, and a central region with neural rosettes with higher cell density and radially arranged nuclei around a central lumen. Specifically, the presence of rosette-like structures composed of proliferative neural stem cells (PAX6^+^/Ki67^+^) organized into polarized tube-like structures surrounded by post-mitotic cortical neurons (SYP^+^) was confirmed.

To further characterize cORGs, total RNA was extracted from the cORGs derived from three control iPSC lines at day 40 of maturation, and a qPCR was performed. Figure 2B shows the relative quantification (RQ) values of each sample, calculated by comparing the constitutive expression levels of parental iPSCs. A general increase in the expression of all the markers compared to the parental iPSCs was observed, except *OCT4*, which, being a marker of pluripotency, was significantly decreased (*p* < 0.0001). *SOX2*, a radial cell marker, showed no significant differences in expression as it is also a key transcription factor in iPSCs, resulting in comparable relative levels between the two populations. *PAX6*, a neural progenitor marker, was the only gene to show a statistically significant increase (*p* < 0.02). Although the expression levels of *TUBB3*, *DCX*, and *MAP2*—markers of corticogenesis—were increased compared to the basal level detected in parental iPSCs, this increment did not reach statistical significance due to high standard deviations. Finally, the expression of *GFAP*, an astrocytic glial marker, was significantly increased (*p* < 0.02), indicating astrocyte differentiation within the cORGs.

The protocol developed in this study leads to the generation of cortical brain organoids with an internal cytoarchitecture resembling the early-stage cortical layer arrangement observed in vivo during neural development. As demonstrated by IHC analyses, typical proliferating neural stem cells are present, surrounded by post-mitotic neurons undergoing maturation. The neural rosettes in the peripheral region appear compact and well structured, exhibiting a highly organized tubular arrangement. In contrast, the central region alternates areas with neurons characterized by small nuclei and extremely elongated and thin axons, and more loosely organized, fragmented areas devoid of SYP^+^ cells, likely indicative of necrosis. The manual Matrigel^®^ embedding, simulating the ECM, supports this differentiation. However, the matrix used, being a xenobiotic ECM derived from mouse sarcomas, exhibits heterogeneity, batch-to-batch variability, and a poorly defined protein composition that can determine the uncontrollability of cellular niches and decrease the reproducibility of organoids [11], as demonstrated by the higher error bars in qPCR. Furthermore, the manual embedding process is both labor-intensive and challenging in terms of maintaining sterility. The inconsistent placement of organoids within the Matrigel^®^ drop can compromise the structural stability of the embedding, leading to poor internal uniformity and compaction. Since each organoid interacts with a different portion of the gelled matrix, these variations may ultimately influence differentiation outcomes and limit standardization.

### 2.3. Hydrogel-Based Ink Stability and Permeability Evaluation

The highest challenge in using brain organoids for disease modeling is the variability in size and morphology between different organoid cultures, especially due to labor labor-intensive manual processes and dependence on technician skills. Achieving reproducibility requires process standardization, where bioengineering and automation play a crucial role. In the present study, we describe a novel method for generating cORGs, which, by its automation and rapidity, and use of only defined media and matrix components, may enhance standardization and reproducibility. A chemically defined hydrogel composed of Matrigel^®^, sodium alginate (SA), and sodium carboxymethylcellulose (CMC) was formulated as a bioink for the 3D bioprinting of nEBs. Before bioprinting, extensive testing was performed to assess hydrogel stability and permeability. Firstly, degradation tests were performed on the SA-CMC-Matrigel^®^ hydrogel after CaCl_2_ crosslinking. As shown in Figure 3A, the hydrogel tends to gradually lose weight over time when cultured in an incubator at 37 °C, 5% CO_2_ in a neural medium. Specifically, the hydrogel’s weight loss remained contained during the first weeks, while after 1 month in culture under the same conditions, the hydrogel was fully degraded. The hydrogel stability during the first two weeks of iPSC-derived cORG generation is crucial to sustain the neural induction and to guide the internal cytoarchitecture. The gradual weight loss of the hydrogel from day 3 indicates a controlled degradation process, allowing organoids to transition from a scaffold-supported state to a more self-organized structure. This mimics the in vivo extracellular matrix remodeling, facilitating nutrient diffusion, cellular expansion, and network maturation. As the hydrogel degrades, it reduces physical constraints on growing organoids while maintaining enough mechanical support early on, promoting a balance between structural guidance and autonomous tissue development. The proper and robust hydrogel scaffold could simulate the presence of ECM and lead neural stem cells to self-organize and differentiate into radial structures and cortical layers. The correct initial formation of a biological niche would promote a positive feedback loop such that portions of the degraded hydrogel after 4 weeks would be replaced by natural internal neural parenchyma that goes on to sustain the brain organoids’ maturation. At shear rates lower than 5 s^-1^, the SA-CMC-Matrigel^®^ hydrogel exhibits a viscosity of η = 71 ± 5 mPas, which is further lowered to η = 40 ± 0 mPas at 118 s^-1^. (shear thinning effect), as reported in Figure 3B.

### 2.4. Hydrogel-Based Ink Printing Process Characterization

Figure 4A shows the printed samples, indicating that the uniformity and stability of the process sufficed for the obtainment of repeatable 3D-printed structures, which consist of 50 *×* 200 × 200 mm grids. A swelling test evaluated the ability of the SA-CMC-Matrigel^®^ hydrogel-based ink to incorporate water. The hydrogel absorbs liquids with gradually rising kinetics that tend to plateau after 24 h, and it is prone to swelling, ultimately doubling its initial weight, as seen in Figure 4B. Last but not least, a diffusion test using fluorochrome-conjugated bovine serum albumin on the SA-CMC-Matrigel^®^ hydrogel-based ink was evaluated to determine the hydrogels’ ability to permit the interchange of proteins, growth factors, and metabolites. Albumin, a key plasma protein, is often used as a model macromolecule in these tests due to its biological relevance and ability to mimic the behavior of larger molecules or drugs. Diffusion tests on hydrogels are important for evaluating the hydrogel’s potential for biomedical applications, particularly in drug delivery systems, but also in vitro modeling to ensure the correct delivery of the induction factors. Fluorescent microscopy data (Figure 4C,D) confirmed an exponential increase in fluorescence intensity over time, with a 10-fold increase at 180 min compared to t0, confirming the hydrogel’s capacity to release or interact with proteins and other biomolecules.

Overall, the SA-CMC-Matrigel^®^ hydrogel-based ink, formulated, autoclaved, and CaCl_2_ crosslinked, exhibits degradation, viscosity, absorption, and diffusion properties compatible with the long-term culture of brain organoids. The slow degradation rate of hydrogels starts to massively decrease only after three weeks in an incubator at 37 °C, ensuring stability during the first critical weeks supporting proper in vitro cerebral development. The low viscosity favors a homogeneous cell dispersion before printing. Although the post-printing crosslinking agent concentration was used, the final swelling ratio was more than two after 2 h. Moreover, proper exchange of cellular medium, ions, and proteins with dimensions less than or equal to BSA (66 kDa) is guaranteed as revealed by diffusion test results.

### 2.5. Bioprinted Hydrogel-Based Cerebral Organoid Generation

A crucial aspect of 3D bioprinting is selecting materials that closely mimic the native ECM of the target tissue. In the developing brain, ECM consists mainly of glycosaminoglycans (such as hyaluronan, chondroitin sulfate, and heparan sulfate), proteoglycans, glycoproteins like laminins, and smaller amounts of fibrous proteins such as collagen [37]. Several biomaterials have been used in neural tissue engineering, including alginate, agarose, fibrinogen, collagen, and gelatin, all aiming to replicate the specific features of the brain ECM and create more accurate in vitro models for neural regeneration and function [38]. Currently, most 3D bioprinting techniques focus on printing bioinks containing single-cell suspensions [39], which require high cell densities and specific mechanical properties, such as elasticity and viscosity, to support cellular elongation and adhesion. By combining the precise spatial control of 3D bioprinting with the scalability and biological relevance of neural organoids, there is significant potential to develop a more adaptable and reliable model of brain development. To simulate the brain ECM in vitro, we combined the previously described protocol for generating cortical organoids from iPSCs with a 3D bioprinting technique and customized hydrogel-based ink. Specifically, on day 7, a suspension of iPSC-derived nEBs mixed with the SA-CMC-Matrigel^®^ hydrogel-based ink was loaded into a sterile cartridge and printed directly into a 24-well plate. After crosslinking, the 3D-bioprinted cORGs were cultured in a dynamic suspension on an orbital shaker inside an incubator and fed every 3–4 days with maturation media for 40 days (Figure 5A). In contrast to what was observed with the Matrigel^®^-based protocol, the acquisition of typical neural folding by the bioprinted spheroids occurred later (twice the expected time), around day 20 of differentiation. As shown in Figure 5B, till day 10, nEBs appeared spherical with a dense core and well-defined edge, and a diameter of approximately 200 μm. Over time, hydrogel-based 3D-printed cORGs increased in size, and neural rosettes began to appear. By day 40, these organoids reached approximately 2 mm in diameter, matching the size observed at the same time point in Matrigel^®^-based cORGs. The mechanical properties of the hydrogel, such as stiffness, elasticity, and viscosity, play a crucial role in supporting cellular growth, differentiation, and organization. If the hydrogel is too soft or rigid, it may not provide the best mechanical cues necessary for cells to properly align, migrate, and form folds, which are essential for neural development [40]. The stiffness and viscosity of pure Matrigel provide the appropriate mechanical tension to the extracellular matrix, promoting the embedded cells to form tubular structures by arranging themselves radially around a central lumen. Conversely, the hydrogel embedding is less viscous and rigid, which can slow the generation of neural rosettes [41]. Moreover, the slowdown observed in the growth and maturation of hydrogel-based 3D-printed cORGs could also be due to the lower concentration of Matrigel^®^ present in the ink. The literature reports that Matrigel^®^ exposure influences organoid size, morphology, and cell type composition. Particularly, it is reported that the amount of Matrigel^®^ used in the embedding area is directly proportional to the size of the resulting brain organoid [42].

### 2.6. Bioprinted Hydrogel-Based Cerebral Organoid Characterization

The results of morphological and IHC analyses on hydrogel-based 3D-bioprinted cORGs at 40 days of maturation are reported in Figure 6A. The presence of rosette-like structures (PAX6^+^/Ki67^+^) surrounded by cells expressing specific cortical layer markers (SYP^+^) was confirmed. However, in this case, the neural rosettes appeared less organized into clusters and not confined to the peripheral regions, but were more dispersed throughout the organoid, including the central region. The presence of proliferating cells (Ki67^+^) and the increase in organoid size observed over 40 days confirm the biocompatibility of the SA-CMC-Matrigel^®^ hydrogel. Moreover, the qPCR analyses (Figure 6B) verify the statistically significant induction of all the expected neural markers in hydrogel-based 3D-bioprinted cORGs in comparison to parental iPSCs. Despite what was previously described for Matrigel^®^-based cORGs, the induction of the neural rosettes marker *SOX2*, and corticogenesis markers *TUBB3*, *DCX*, and *MAP2* was statistically significant (*SOX2*, *TUBB3*, and *DCX p* < 0.02; *MAP2 p* < 0.001).

### 2.7. Comparison Between Matrigel^®^-Based and Bioprinted Hydrogel-Based Cerebral Organoids

Through double immunofluorescence (IF) staining, we simultaneously observed the expression and localization of PAX6 and SYP protein within the internal cytoarchitecture of cORGs, enabling a comparison between Matrigel^®^-based and hydrogel-based 3D-bioprinted cORGs. As shown in Figure 7, it was confirmed that Matrigel^®^-based cORGs exhibited a more structured and organized arrangement, with PAX6^+^ neural stem cells radially surrounding a central lumen. In contrast, hydrogel-based 3D-bioprinted cORGs displayed a more dispersed cellular distribution throughout the entire organoid, surrounded by SYP^+^ neurons. Despite the presence of neural buds and rosettes in the hydrogel-based 3D-bioprinted cORGs, their overall macroscopic and microscopic definition appeared less distinct. Indeed, the IF analysis revealed that the internal cytoarchitecture is more compact and less fragmented, with SYP^+^ staining covering the entire structure, showing a compact neural network of small neurons. The inclusion of the SA-CMC-Matrigel^®^ hydrogel seems to improve the compactness and organization of cORG structures. As the hydrogel mimics the physical properties of the extracellular matrix, it provides a more conducive environment for cell–cell interaction, proliferation, and self-organization into multilayered structures, offering advantages over traditional flat surface cultures [43].

### 2.8. Early Brain Regionalization Evaluation in Matrigel^®^-Based and Bioprinted Hydrogel-Based Cerebral Organoids

Brain development in vivo exhibits a striking degree of heterogeneity and regionalization as well as interdependence among different brain regions [44]. To test for early brain regionalization in whole organoids, a qPCR for forebrain (*FOXG1*), hindbrain (*EGR2*), hippocampus (*FZD9*), cortical plate (*BCL11B*), dorsal cortex (*EMX1*), choroid plexus (*TTR*), and cortical plate (*TBR1*) gene expression at day 40 of the Matrigel^®^-based cORGs, hydrogel-based 3D-bioprinted cORGs, and parental iPSCs was assessed (Figure 8). The forebrain marker *FOXG1* was significantly upregulated in both organoid types (*p* < 0.01) compared to iPSCs, while *EMX1* (dorsal cortex) showed significant induction only in Matrigel^®^-based cORGs (*p* < 0.02). The gene expression of *TBR1*, cortical marker; *TTR*, choroid plexus marker; and *BCL11B*, cortical plate marker was significantly elevated in both organoid types compared to parental iPSCs (*p* < 0.02). The gene expression induction of *FZD9*, hippocampus marker, was not statistically significant in both organoids in comparison to parental iPSCs. Significant differences between the two types of organoids were observed only in the induction of the hindbrain and dorsal cortex. A strong gene expression reduction in EGR2 was observed in Matrigel^®^-based cORGs. This trend was already described by Lancaster [10,33] as reminiscent of the developmental expansion of forebrain tissue during human brain development. Specifically, in iPSC-derived cORG generated using Matrigel^®^ embedding, FOXG1 expression remains high after 20 days, while hindbrain marker EGR2 tends to decrease. Moreover, he also demonstrated that the forebrain region displayed typical cerebral cortical morphology strongly positive for EMX1, indicating dorsal cortical identity [10,33]. Thus, the observed pattern of high expression of *EGR2* and *EMX1* in our Matrigel^®^-based cORGs is consistent with what is described in the protocol developed by Lancaster. In contrast, it is important to note that this trend is opposite in our 3D-bioprinted hydrogel-based cORGs, which at day 40 of maturation showed higher gene expression of *EGR2* and lower gene expression of *EMX1* in comparison to Matrigel^®^-based cORGs. This may reflect a delay in the hydrogel-embedded bioprinted cORGs’ maturation. This hypothesis is also supported by the morphological evidence that bioprinted nEBs in the hydrogel began to form buds and folds about ten days later than Matrigel^®^ embedding nEBs.

Finally, both cORG types showed a high expression of *TTR*, a marker of a specific brain structure, namely the choroid plexus (ChP). Despite this, it is important to note that in hydrogel-based cORGs, the *TTR* expression is about 50% less than in Matrigel^®^-based cORGs. These data are coherent with that already reported [42] about greater amounts of Matrigel^®^ that can promote an increase in the number of ChP cells through the *BMP4* pathway. ChP is a highly vascularized secretory tissue located within each ventricle of the vertebrate brain. It plays a crucial role in supporting the central nervous system by producing up to 500 mL of cerebrospinal fluid (CSF) daily in an adult human brain. During development, ChP also produces various growth factors and cytokines that regulate cortical development, neurogenesis, and the immune system. [45]. The presence of typical brain structures, such as ChP, enhances the in vitro model’s physiological relevance, making bioprinted cORGs a promising tool for studying different neurological pathologies, including neuroinflammatory ones associated with CSF alterations [46].

## 3. Conclusions

The definition of a tissue organoid continues to evolve, but a recent proposal highlights key characteristics: the presence of multiple organ-specific cell types and the organization of cells and structures in a way that reflects the spatial architecture of the organ. In this manuscript, we showed that the SA-CMC-Matrigel^®^ hydrogel can sustain the generation of iPSC-derived cortical cerebral organoids as the conventional Matrigel^®^-based approach does. By day 40 of differentiation, the organoids had developed into large (up to about 2 mm in diameter), heterogeneous, and consistent masses, with a cytoarchitecture resembling an early-stage developmental fetal brain composed of densely packed cell somas, with extensive neurites SYP^+^. Among these were neurite bundles and radiating neurites that formed from regions of neural progenitor cells PAX6^+^/Ki67^+^ that arranged into tubular structures. Moreover, neural stem/progenitor cell marker PAX6 showed persistent neurogenesis, *MAP2/DCX* expression suggested the formation of cortical tissue-like neuronal layers, and molecular characterization of strong *FOXG1* expression and cortical plate (*TBR1*) marker expression demonstrated dorsal forebrain identity. The advancement of bioinks for bioprinting neural organoids is crucial and has garnered increasing attention in recent years. Biomaterials chosen for hydrogel composition are seldom biologically neutral. They can transmit or receive biological signals to and from cells, offer adhesion sites for cells, and shape cellular microenvironments. Many hydrogels, such as native collagen, fibrinogen, and Matrigel, which have good biological qualities for maintaining brain organoid development, lack the physicochemical qualities needed for 3D bioprinting. Pneumatic extrusion cannot be used to extrude Matrigel^®^, an ECM protein combination that contains over 1800 distinct proteins, including laminin (55%), collagen IV (30%), and entactin (6%), since it acts erratically under pressure and ejects from the syringe in an unpredictable manner [12,38]. To address this issue, some research groups crosslinked Matrigel^®^ with collagen or polyethylene glycol (PEG), but those composite hydrogels resulted in a dense, poorly organized structure, leading to highly heterogeneous scaffolds [47]. Single polymers address some of the challenges associated with matrix extracts. Agarose and alginate are two examples of natural polymers that are frequently employed to support the 3D cultivation of neuronal cells, tissues, or organoids. Using an agarose-based bioink in conjunction with alginate, a polymer that is far more widely used than agarose, one study [48] showed that it was feasible to create functioning brain tissue. According to studies, 1% alginate fosters neural development and differentiation, has an elastic modulus similar to brain tissue, and structurally mimics hyaluronic acid [49]. A breakthrough in 2021 successfully generated functional dopaminergic neurons from human iPSCs in alginate beads, maintaining their function for up to 50 days without necrosis [50]. However, alginate-based hydrogels face two key issues: poor cell adhesion and limited long-term outcomes in culture medium due to the replacement of divalent cations with monovalent ones. To address these challenges, alginate hydrogels are often supplemented with additional agents, such as fibronectin, to improve stability and enhance cell adhesion [51]. A recent interesting work published by Tomaskovic-Crook and colleagues reported the use of Gelatin Methacryloyl (GelMA) hydrogel to generate brain organoids displaying dorsal forebrain identity and characteristic cortical tissue architecture, with cells that arranged to form tuboid structures [52]. GelMA is a derivative of gelatin, a natural polymer obtained from collagen, that has been modified by the addition of methacryloyl groups. The major disadvantage of using GelMA is that the photoinitiated crosslinking process can introduce cytotoxicity due to DNA damage. High doses of UV light, for instance, may harm sensitive neural cells, affecting organoid viability. To date, a valid alternative to ECM-embedded cORGs is microfluidic-based approaches. This system uses precisely engineered channels that allow for better nutrient and waste management through active fluid flow, which improves the growth and development of organoids. Such platforms can integrate more complex environmental conditions, like mechanical forces, oxygen gradients, and controlled application of signaling molecules, providing a more physiologically relevant model of brain development [53]. Automation in microfluidic systems significantly reduces labor intensity compared to manual methods, as processes like nutrient delivery and media changes can be automated, leading to more efficient and scalable experiments [54]. However, microfluidic systems are more expensive and technically challenging to set up, requiring specialized equipment and expertise. For studies that require high control, reproducibility, and scalability, especially in disease modeling or drug screening, microfluidic-based approaches may be the best option. However, for smaller-scale studies or when cost is a major factor, manually Matrigel^®^-embedded organoids may be sufficient, offering a simpler but still valuable platform for investigating brain development and cellular behavior. In terms of biological relevance, Matrigel embedding being more affordable, becomes more widespread, which can promote comparison between different laboratories. Based on the above, it is evident that the search for the perfect biological niche for iPSC-derived brain organoid in vitro generation is still ongoing.

Overall, this study aimed to highlight the labor-intensive nature of the conventional protocol for generating iPSC-derived brain organoids, which involves manual embedding in pure Matrigel^®^, and not to investigate alternative materials to Matrigel^®^, which is known to pose reproducibility issues. Our protocol aims to eliminate the operator-dependent manual procedure by leveraging 3D bioprinting technology, which enables the automation and standardization of the method. Bioprinting allows the precise deposition of bioinks to form a 3D structure following a direct printing technique of cell-laden materials, different from traditional molding. To our knowledge, to date in the literature, the proposed bioink, composed of SA-CMC combined with a ten-fold-diluted Matrigel^®^, has not been described yet. The SA-CMC-Matrigel^®^ hydrogel is promising for the 3D bioprinting of cerebral organoids, effectively reducing material costs and variability due to manual process embedding, and minimizing batch-to-batch variability. This formulation, still containing Matrigel^®^, albeit in reduced quantities, retains a substantial number of proteins, ligands, and growth factors abundant in in vivo ECMs, which are essential to maintain the bioink’s functional properties and can be sufficiently diffused throughout the bioink itself as demonstrated here. Moreover, bioprinting can be automated to eliminate the high variation resulting from the operator, the laboratory equipment, the materials batch, and operating conditions. From this study, a more stable and uniform process can be identified to promote access to similar techniques and biological outcomes for different researchers and research units. The main limitations of this work are relying on the possible outcomes regarding organoids’ function and response, which are yet to be characterized by the authors, and the mechanical stability over time of the bioink used here, which has to be enriched further to be versatile and customizable for diverse neural applications.

## 4. Materials and Methods

### 4.1. Stem Cell Culture

A commercial human iPSC control cell line: Gibco™ episomal hiPSC line (cat. n. A18945, ThermoFisher Scientifc, Inc., Waltham, MA, USA) from cord-blood CD34+progenitors’ cells and a control iPS cell line previously reprogrammed at our lab from a commercial cell line of fibroblasts BJ (ATTC CRL-2522) were used. hiPSCs were verified to be pluripotent, mycoplasma free (Appendix A), and fed daily with NutriStem^®^ hPSC XF Medium (Sartorius AG, Göttingen, Germany) with the addition of 10 ng/mL of bFGF (Basic fibroblast growth factor; Miltenyi Biotec GmbH., Bergisch Gladbach, Germany), manually picked every 4–5 days on new Matrigel^®^ (Corning Inc., Corning, NY, USA)-coated well plates and cultured at 37 °C in 5% CO_2_ as previously described [55].

### 4.2. Cerebral Organoid Generation and Culture Conditions

Cerebral organoids were generated using commercial media enriched with growth factors and supplements (Cerebral Organoid kit, catalog #08570, #08571, Stem Cell Technologies, Vancouver, Canada). The process is divided into four steps, each of which has a designated medium: hiPSC-derived embryoid body (EB) generation, neuroectodermal induction, matrix embedding for the neuroepithelium expansion, and cerebral organoid maturation. Briefly, hiPSCs at 70/80% of confluency were dissociated to single-cell suspension and seeded in Organoid Formation Medium into AggreWell™ Microwell Plates (StemCell Technologies) following the manufacturer’s instructions to obtain a scalable production of neural embryoid bodies (nEBs) similar in shape and size (6 × 10^3^ cells/nEB). The day after, NEBs were transferred to ultra-low adherent 6-well plates (Corning Inc., USA) for floating culture in the same medium for 5 days. Then, the medium was switched to an Induction Medium to promote the neural ectodermal expansion. On day 7, at least 10 nEBs were singly embedded in matrix Matrigel^®^: using a wide-bore 200 µL pipette tip, each nEB was collected with a small volume of medium and transferred to the embedding surface (sheets of dimpled parafilm). After removing the excess medium using a pipette with a cold 200 µL standard pipette tip, 15 µL of matrix Matrigel^®^ were dispensed dropwise onto each nEB, repositioning it in the center of the droplet and incubating for 20 min at 37 °C to gel. Thus, the sheet with embedded organoids was positioned by sterile tweezers directly above one well of a 6-well ultra-low adherent plate and was gently washed with an Expansion Medium to let the droplets fall off the sheet and into the well, and the plate was incubated at 37 °C for 3 days. Finally, from day 10 to 30, cerebral organoids were moved to an incubator with an orbital shaker and were fed every 3–4 days with maturation media.

### 4.3. Immunohistochemistry and Immunofluorescence

Microscopic analysis: Organoids were formalin-fixed and paraffin-embedded with a standard protocol. Two-micron-thick tissue sections were H&E-stained and checked for cell content. For immunohistochemistry, the sections were deparaffinized and rehydrated in graded solutions of ethanol and distilled water. Endogenous peroxidase was blocked by incubation with methanol and hydrogen peroxide 0.03% for 20 min during rehydration. Immunostaining was performed using anti-Sinaptofisin (SYP, clone DAK-SYNAPT, ready to use, Agilent, Santa Clara, California, USA, anti-PAX6 (clone EP341, 1:200,Epitomics, Inc. - Burlingame, USA), and anti-Ki67 (clone 30-9, ready to use, Ventana Medical Systems, Oro Valley, USA ). After pretreatment with microwave in the EDTA buffer at pH 8.00 (2 cycles of 5 min at 1000 watts and 3 cycles of 5 min at 750 watts), the reaction was revealed using Novolink Polymer (Leica Microsistem, Wetzlar, Germany), followed by diaminobenzidine (DAB, Dako, Glostrup, Denmark). Finally, the slides were counterstained with Meyer’s Haematoxylin. For the immunofluorescence, the polymer amplifications were followed by tyramide fluorophore AF488 (Thermofisher, Waltham, Massachusetts, USA #B40953) and CF647 (Biotium, Fremont, CA #96022). After completing the first immune reaction, the second was visualized using the different tyramide fluorophores. VECTASHIELD (Antifade Mounting Medium with DAPI, H-1200, Vector Lab, Newark, CA ) was used to counterstain and cover the tissue fluorescent slides. Sections were digitized using Axioscan7 (Zeiss, Jena, Germany) at 20× magnification. For immunofluorescence slides, the Colibri 7 lamp was used as an LED light source with different illumination wavelengths (450–488; 615–648) depending on the different modules of the LED (385 nm, 475 nm, and 630 nm).

### 4.4. RNA Extraction and qPCR

Total RNA was extracted using the NucleoSpin^®^ RNA II kit (Macherey–Nagel, Düren, Germany). The ImPromIITM Reverse Transcription System (Promega Corporation, Madison, WI, USA) was used to retrotranscribe 500 µg of total RNA. SYBR Green was used to perform the qPCR gene expression analysis (Bio-Rad Laboratories, Inc., Hercules, CA, USA). Table 1 lists the primer pairs (IDT, Inc., Coralville, IA, USA). The thermocycler conditions were 98 °C for 30 s, 39 cycles of 95 °C for 5 s and 60 °C for 30 s, followed by 65 °C for 5 s. The CFX96 C1000 TouchTM Real-Time PCR Detection System (Bio-Rad Laboratories)was used to perform the assays, and Bio-Rad Laboratories, Inc.’s CFX Manager software v.3.1 was used to assess the results. The 2^−ΔΔCT^ method was used to calculate the relative quantification of the target genes, using *βACTIN* and *HSP90* as reference genes. Data were determined from 3 independent experiments.

### 4.5. Hydrogel Bio-Ink Characterization and 3D Printing Parameters

The hydrogel was developed by dissolving 1.5 *w*/*v*% SA (low viscosity, Thermo Fisher Scientific, Waltham, MA, USA) and 1 *w*/*v*% of NaCMC (Mw = 250 kDa, Thermo Fisher Scientific) in Milli-Q water, resulting from different hydrogel formulations demonstrated by the authors [56]. The solution was then sterilized in an autoclave at 121 °C for 20 min. When the mixture reached a warm temperature, a proper volume (1:10 dilution) of Neurobasal Medium (Thermo Fisher Scientific) containing hiPSC-derived nEBs at day 7 of generation and matrix Matrigel (1:10) were added. Printing was conducted in a 12-well plate with the 3D-Bioplotter system (EnvisionTEC, DE, Dearborn, MI, USA). The experimental parameters used for the extrusion printing were a pressure of 0.4 bar, a speed of 100 mm/s, and a needle diameter of 0.6 mm as already described in previous works from the authors [57]. In total, 0.5 mL of 1 *w*/*w*% CaCl_2_ was used for crosslinking on the top of each hydrogel and kept for 20 min in an incubator at 37 °C. Once crosslinking was completed, hydrogels were washed twice in PBS 1X, covered with the Expansion Medium, and placed in an incubator for the next three days. Finally, the medium was switched to a maturation medium, with feeding hydrogel every 3–4 days.

Before printing nEBs, a layer of the same ink without cells (thickness of 2 mm) was printed into each well of a 12-well plate, crosslinked with 1 *w*/*w*% CaCl_2_ for 20 min at room temperature, washed twice with PBS 1X and stored in the fridge till use. This printed layer was poured into the well to avoid the direct contact of nEBs printed later to the plastic surface of the well and the consequential adhesion in a monolayer.

Viscosity: The viscosity of the hydrogel was measured using a VISCOTM 6800 (Atago Co., Ltd., Tokyo, Japan) viscometer, and data were recorded using the Tera Term software, version 5.4.0 (Tera Term Project). For each velocity applied, at least 15 data points were acquired. The viscometer measures a viscosity value η [mPa·s] at each rotation. Standard deviations were computed for each speed value. Based on the beaker’s inner diameter (DB), the spindle’s external diameter (DS), and the angular speed in radians (*ω*), the shear rate is determined using Equation (1): (1)$$Shear\;rate\;[s^{-1}]=2\;\frac{D_{B}^{2}}{D_{B}^{2}–D_{S}^{2}}\cdot \omega \;$$

Degradation tests: The SA-CMC-Matrigel^®^ hydrogel-based inks without cells, after crosslinking, were washed twice with PBS 1X and placed in a 33 mm Petri dish, filled with complete neurobasal medium and incubated at 37 °C with 5% CO_2_. At specific time points (1, 3, 7, 14, 21, and 30 days), the samples were weighed (Wt) after removing the medium and were dehydrated at 40 °C overnight in a thermal oven. The dry weight (Wd) of the samples was recorded. Finally, the degradation rate of each sample was calculated as the difference between Wt and Wd.

Diffusion rate test: The SA-CMC-Matrigel^®^ hydrogel-based inks without cells, after crosslinking, were washed twice with PBS 1X, placed in a 33 mm Petri dish, weighed, and filled with complete neurobasal medium with fluorochrome-conjugated bovine serum albumin (BSA, excitation/emission 593/614, Biotium, San Francisco, CA, USA), 10 µg/mL, in the dark. At every time point (1, 15, 30, 60, and 180 min), the hydrogel films were washed with Milli-Q water, moved into a new 12-well plate, and their fluorescence was acquired with the fluorescence microscope Olympus IX70 (Olympus, Tokyo, Japan) at 10× magnification. Images were collected at 700 ms exposure with the software Image-Pro Plus v7.0 (Media Cybernetics, Rockville, MD, USA). The fluorescence intensity was analyzed with the software ImageJ, version 1.47t (NIH, Bethesda, Maryland, USA), and the relative intensity was calculated by normalizing the fluorescence at each time point to t0 (without fluorochrome-conjugated BSA).

Swelling ratio test: The SA-CMC-Matrigel^®^ hydrogel-based inks without cells, after crosslinking, were washed twice with PBS 1X, placed in a 33 mm Petri dish, weighed (Wt0), and filled with 3 mL of complete neurobasal medium. Relative weights (Wt) were measured at the time points 30, 60, 120 min, 24 h, 48 h, and 7 days. For each weighing, the medium was removed, and the hydrogel was placed on a filter paper for 10 s to absorb any excess medium culture and then weighed. The swelling index was calculated as the difference between Wt0 and Wt. All these tests were carried out with 3 replicates for each measurement.

### 4.6. Statistical Analysis

The experiments were repeated at least three times. The data were expressed as mean ± standard deviation. Comparisons between groups were performed using Student’s *t*-test on the GraphPad Prism 8 software (San Diego, CA, USA).

## Figures and Tables

**Figure 1 gels-11-00284-f001:**
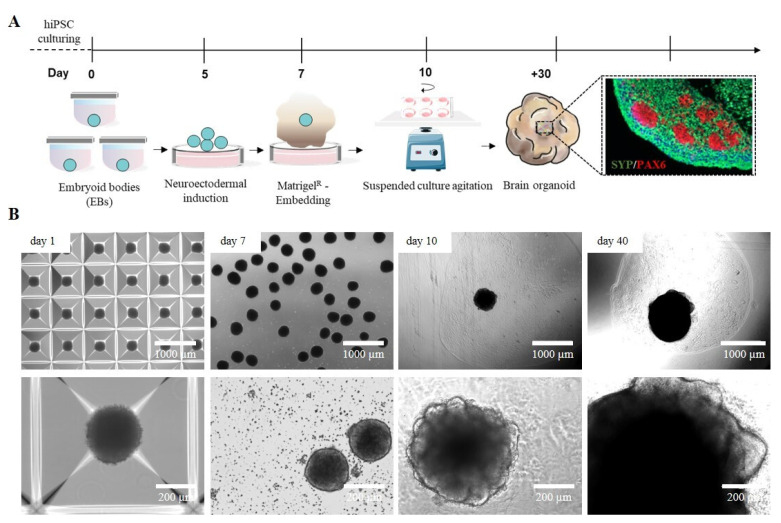
(**A**) Schematic workflow of the Matrigel^®^-embedded cORGs. (**B**) Representative images of the main steps of iPSC-derived cORG formation.

**Figure 2 gels-11-00284-f002:**
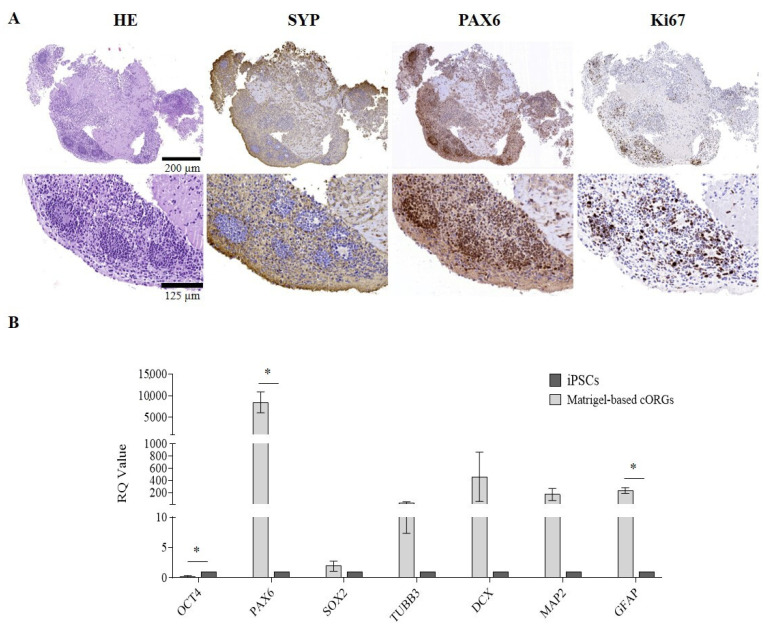
(**A**) Matrigel^®^-embedded cORGs microscopic analysis: Hematoxylin and Eosin (H&E), synaptophysin (SYP), paired box 6 (PAX6), and Ki67. Top: magnification 14×, scale bar 200 µm; center: magnification 45×, scale bar 125 µm. (**B**) Quantitative PCR assay for *OCT4*, *PAX6*, *SOX2*, *TUBB3*, *DCX*, *MAP2*, and *GFAP* expression in Matrigel^®^-embedded cORGs at day 40. Data were normalized on *βACTIN* and *HSP90AB1*, and calculated for the parental iPSC line. The bars represent the mean ± standard deviation (n = 3: BJ hiPSC line n = 2 and episomal hiPSC line n = 1). Student’s *t*-test, * *p* < 0.05.

**Figure 3 gels-11-00284-f003:**
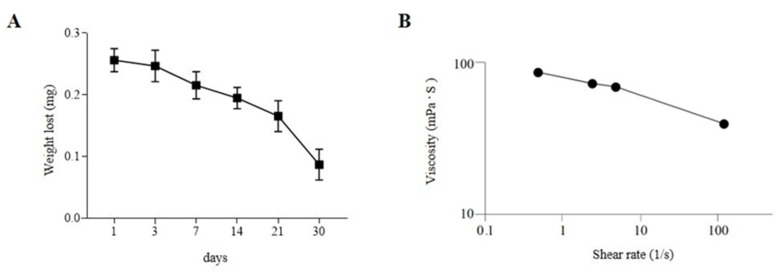
(**A**) Degradation tests performed on SA-CMC-Matrigel^®^ hydrogel. (**B**) Viscosity tests performed on SA-CMC-Matrigel^®^ hydrogel, expressed as shear rate function.

**Figure 4 gels-11-00284-f004:**
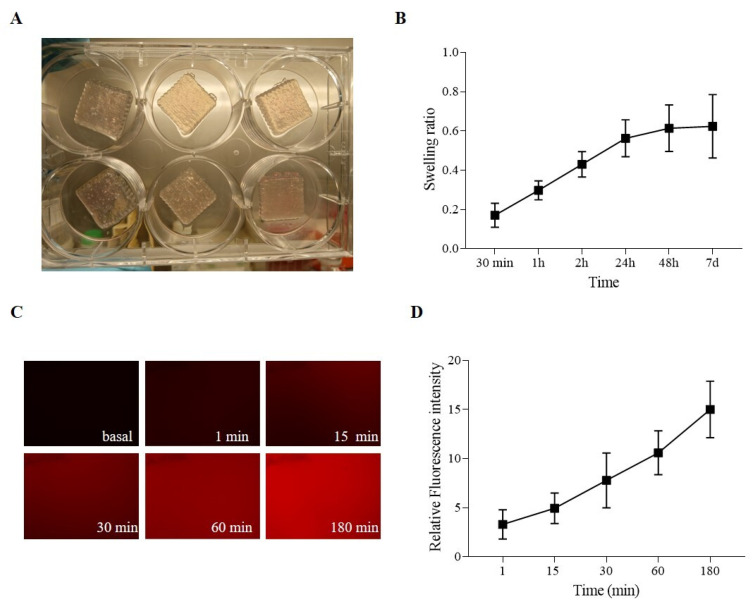
(**A**) SA-CMC-Matrigel^®^ hydrogel 3D-printed samples. (**B**) Swelling ratio. (**C**) Diffusion tests: 700 ms exposure, 10× magnification, and fluorescence pictures of the bioinks at various times. (**D**) Diffusion tests: fluorescence intensity graph derived from normalized fluorescence pictures on materials that were not exposed to BSA.

**Figure 5 gels-11-00284-f005:**
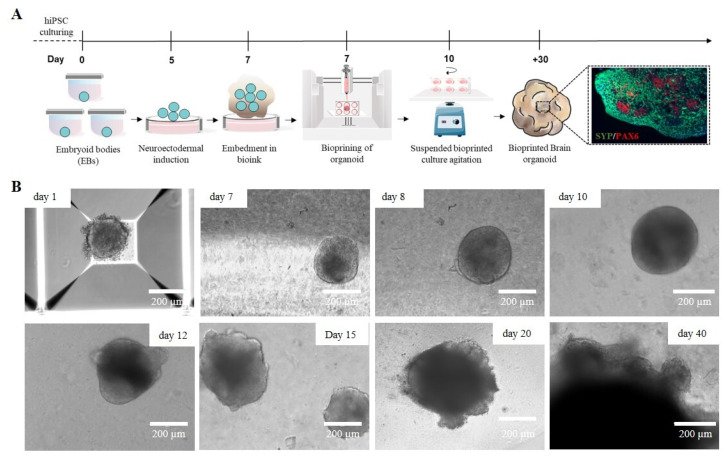
(**A**) Schematic workflow of the 3D-bioprinted hydrogel-based cORGs. (**B**) Representative images of the main steps of iPSC-derived cORG formation.

**Figure 6 gels-11-00284-f006:**
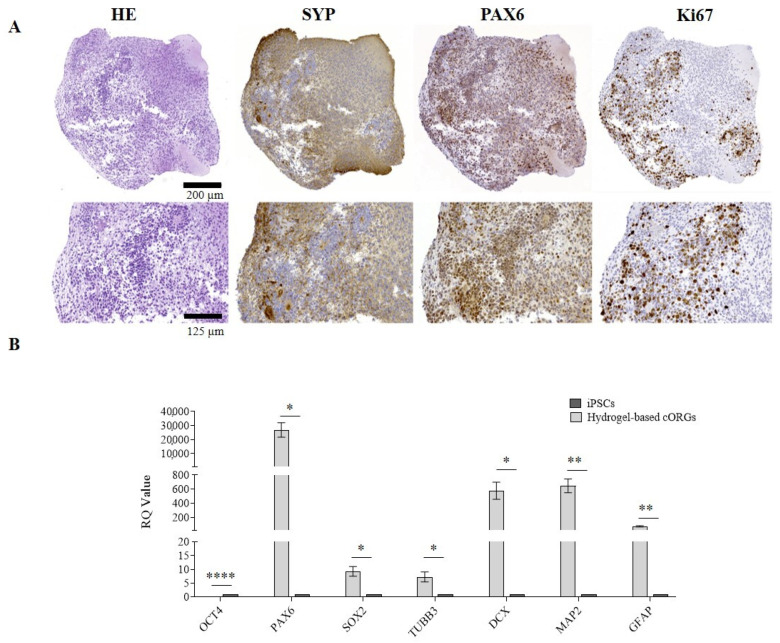
(**A**) Microscopic analysis: Hematoxylin and Eosin (H&E), synaptophysin (SYP), paired box 6 (PAX6), and Ki67. Top: magnification 18×, scale bar 200 μm; bottom: magnification 45×, scale bar 125 μm. (**B**) Quantitative PCR assay for *OCT4*, *PAX6*, *SOX2*, *TUBB3*, *DCX*, *MAP2*, and *GFAP* expression in Matrigel^®^-based cORGs at day 40. Data were normalized on *βACTIN* and *HSP90AB1*, and calculated for the parental iPSC line (black bars). The bars represent the mean ± standard deviation (n = 3: BJ hiPSC line n = 2 and episomal hiPSC line n = 1). Statistical analyses were performed using Student’s *t*-test, * *p* < 0.05, ** *p* < 0.01, and **** *p* < 0.0001.

**Figure 7 gels-11-00284-f007:**
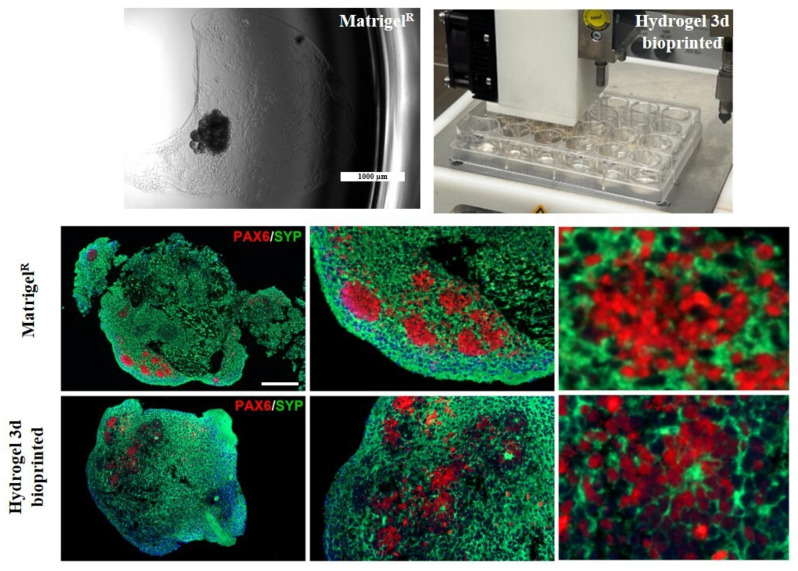
Representative images for Matrigel^®^-based and 3D-bioprinted cORGs and immunofluorescence assay: double staining of synaptophysin (SYP) in green, and paired box 6 (PAX6) in red. Nuclei are stained with DAPI in blue. LEFT: magnification 18×, scale bar 200 µm; center: magnification 45×, scale bar 125 µm; right: magnification 150×, scale bar 40 µm.

**Figure 8 gels-11-00284-f008:**
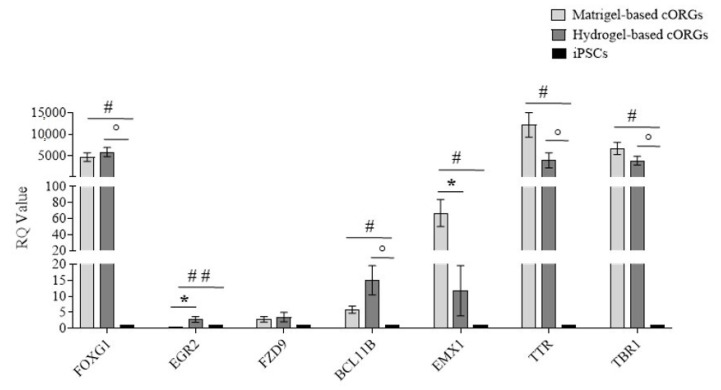
Quantitative PCR assay for *FOXG1*, *EGR2*, *FZD9*, *BCL11B*, *EMX1*, *TTR*, and *TBR1* expression in Matrigel^®^-based and 3D-bioprinted cORGs at day 40. Data were normalized on *βACTIN* and *HSP90AB1* and calculated for the parental iPSC line (black bars). The bars represent the mean ± standard deviation (n = 3: BJ hiPSC line n = 2 and episomal hiPSC line n = 1). Statistical analysis was performed using Student’s *t*-test. * Matrigel^®^-based cORGs vs. hydrogel-based cORGs, # * Matrigel^®^-based cORGs vs. iPSCs, ° hydrogel-based cORGs vs. iPSCs. *, #, and ° *p* < 0.05, ## *p* < 0.01.

**Table 1 gels-11-00284-t001:** Primer set used for cORG characterization.

Gene	Forward 5’-3’	Reverse 5’-3’
OCT4	CCTCACTTCACTGCACTGTA	CAGGTTTTCTTTCCCTAGCT
PAX6	CTGAAGCGGAAGCTGCAAAG	TTGCTGGCCTGTCTTCTCTG
SOX2	CCCAGCAGACTTCACATGT	CCTCCCATTTCCCTCGTTTT
TUBB3	GGCCAAGTTCTGGGAAGTCAT	CTCGAGGCACGTACTTGTGA
DCX	TATGCGCCGAAGCAAGTCTC	TACAGGTCCTTGTGCTTCCG
MAP2	GACTGCAGCTCTGCCTTTAG	AAGTAAATCTTCCTCCACTGTGAC
GFAP	GAGGTTGAGAGGGACAATCTGG	GTGGCTTCATCTGCTTCCTGTC
FOXG1	ACAGCTCCGTGTTGACTCAG	AGGGGTTGAGGGAGTAGGTC
EGR2	TTGACCAGATGAACGGAGTG	CTTGCCCATGTAAGTGAAGGT
FZD9	GCGAGAACCCCGAGAAGTT	GTGAAGGCGGTGGAGAAGAA
BCL11B	GCCAGTGTCAGTTGTCAGGT	AGGTTGAAGGGGTTGCTGTC
EMX1	GAGACGCAGGTGAAGGTGTG	CTCGTGGGTTTGTGGTTGC
TTR	TGGCTTCTCATCGTCTGCTC	CGGAGTCGTTGGCTGTGAAT
TBR1	ACAATGGGCAGATGGTGGTT	TGACGGCGATGAACTGAGTC
βACTIN	CGCCGCCAGCTCACCATG	CACGATGGAGGGGAAGACGG
HSP90AB1	TCCGGCGCAGTGTTGGGAC	TCCATGGTGCACTTCCTCAGGC
NANOG	TGAACCTCAGCTACAAACAG	TGGTGGTAGGAAGAGTAAAG
OCT4	CCTCACTTCACTGCACTGTA	CAGGTTTTCTTTCCCTAGCT
SOX2	CCCAGCAGACTTCACATGT	CCTCCCATTTCCCTCGTTTT
KLF4	GATGAACTGACCAGGCACTA	GTGGGTCATATCCACTGTCT
cMYC	TGCCTCAAATTGGACTTTGG	GATTGAAATTCTGTGTAACTGC
Mycoplasma detection	ACTCCTACGGGAGGCAGCAGTA	TGCACCATCTGTCACTCTGTTAACCTC

## Data Availability

The original contributions presented in this study are included in the article/Appendix A. Further inquiries can be directed to the corresponding author.

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
