# Peer review of "Three-Dimensional-Bioprinted Embedded-Based Cerebral Organoids: An Alternative Approach for Mini-Brain In Vitro Modeling Beyond Conventional Generation Methods"

_gels, 2025, doi:10.3390/gels11040284_

Round 1
Reviewer 1 Report
Comments and Suggestions for Authors
The manuscript reports the results of creating cerebral organoids using 3D printing vs manual Matrigel formation. Nevertheless, it is strongly recommended to significantly improve the description of some results and the discussion. In addition, some details related to the introduction, methods, and results might be included to increase the robustness of the study.
- In the introduction, lines 50-52, clarify which are the conditions to create a complex 3D tissue structure.
- Lines 60-61, what is the term complex genetics referred to?
- Lines 101-103, explain in more detail or include some of the ECM components that are absent as well as the mechanical properties.
- How was decided to use sodium alginate (SA), and sodium 260 carboxymethylcellulose (CMC) to prepare the ink for fabricating the hydrogels?
- The authors mention that the hydrogel loses weight in time; nevertheless, it is not clear whether this lost, which is noticed from day 3 (Figure 2A), would be beneficial for the organoids.
- Explain in more detail why the hydrogel takes 7 days for reaching its maximum swelling (Figure 2D). A hydrogel should be completely swollen after 24 hours.
- The authors mention that the maturation time for the 3D printed organoid is longer compared to the manual Matrigel organoid. Please, explain why the 3D organoids were not matured for more time.
- Include qPCR results for different timepoints and not only after 40 days of maturation.
- Include the pluripotent markers for the iPS cell line previously reprogrammed. Add this information as supplemental data.
- In the results and discussion part, it is recommended to include other articles and compared the obtained results with them. The manuscript lacks discussion of the results by comparing them with published literature.
- The manuscript is well-written; nevertheless, it is recommended to check the use of abbreviations as well as the use of uppercases. In some parts, connections between ideas need to be improved.
Comments on the Quality of English Language
The manuscript is well-written; nevertheless, it is recommended to check the use of abbreviations as well as the use of uppercases. In some parts, connections between ideas need to be improved.
Author Response
The manuscript reports the results of creating cerebral organoids using 3D printing vs manual Matrigel formation. Nevertheless, it is strongly recommended to significantly improve the description of some results and the discussion. In addition, some details related to the introduction, methods, and results might be included to increase the robustness of the study.
The authors thank the Reviewer for accepting to read and review the paper.
Comments:
- In the introduction, lines 50-52, clarify which are the conditions to create a complex 3D tissue structure.
The authors thank for the comment and modify the text accordingly:
“These cells, such induced pluripotent stem cells (iPSCs) or embryonic stem cells (ESCs) demonstrate the abilities to self-renew and to self-organize under various conditions to form complex 3D tissue structures called organoids, with advanced morphological and functional fidelity to the in vivo counterpart”.
in
“These cells, such as induced pluripotent stem cells (iPSCs) or embryonic stem cells (ESCs), demonstrate the abilities to self-renew and to self-organize under various conditions to form complex 3D tissue structures called organoids, with advanced morphological and functional fidelity to the in vivo counterpart. Inducing stem cells to differentiate into 3D organoids involves a series of well-defined conditions that mimic the in vivo develop-mental environment. These conditions vary depending on the type of organoid being gen-erated (e.g., brain, liver, gut, kidney) and usually involve specific growth factors or cyto-kines, determinants coated surfaces of adhesion, and mechanical and physical factors such as oxygen levels, shear stress, temperature, and pH”.
- Lines 60-61, what is the term complex genetics referred to?
The authors clarify this point modifying the text accordingly:
“Especially for tissues with difficulties in having a biopsy, such as the nervous tissue, there is a particular need for the development of iPSCs-derived in vitro disease model that can fill the gap and lead the study of neurological disorders with complex genetics”.
In
“Especially for tissues with difficulties in having a biopsy, such as the nervous tissue, there is a particular need for the development of iPSCs-derived in vitro disease model that can fill the gap and lead the study of neurological disorders that can present with a variety of cognitive, social, and motor impairments, and a multifactorial genetic“.
- Lines 101-103, explain in more detail or include some of the ECM components that are absent as well as the mechanical properties.
The authors modify the text accordingly:
“Moreover, Matrigel®, does not fully mimic the complex and dynamic nature of the human brain's extracellular environment, which can limit the physiological relevance of brain organoids in disease modeling [14]. For example, the lack of certain human-specific ECM components or the absence of controlled mechanical properties may impact the development and maturation of brain organoids. “
in
“Moreover, Matrigel® does not fully mimic the complex and dynamic nature of the human brain's extracellular environment, which can limit the physiological relevance of brain organoids in disease modeling [14]. For example, the absence or low levels of hu-man-specific ECM proteins, such as tenascin-C, fibronectin, and laminin isoforms, partic-ularly laminin-511 and laminin-521, and the lack of human-specific proteoglycans like brevican, neurocan, and versican, and heparan sulfate proteoglycans. Moreover, Matrigel shows batch-to-batch variability in stiffness, which affects reproducibility in brain organ-oid cultures. It also lacks tunability (elastic modulus (E) typically ranges from 100 to 400 Pa), which is softer than the developing human brain ECM (~500–1000 Pa), potentially limiting the proper mechano-transduction cues for neural differentiation”.
- How was decided to use sodium alginate (SA), and sodium 260 carboxymethylcellulose (CMC) to prepare the ink for fabricating the hydrogels?
Sodium alginate (SA) and carboxymethylcellulose (CMC) hydrogels offer a more controllable and biomimetic alternative to Matrigel® for brain organoid culture. SA hydrogels can be precisely adjusted in stiffness through ionic crosslinking, better matching the elasticity of human brain ECM. CMC enhances viscosity and printability, ensuring structural integrity during fabrication. Beyond mechanics, SA-CMC hydrogels can be chemically functionalized with human-relevant ECM proteins. This customization supports neural stem cell adhesion, differentiation, and network formation while providing a stable and reproducible environment for long-term studies. Additionally, SA-CMC hydrogels are compatible with 3D bioprinting, enabling the creation of spatially defined brain microenvironments providing structural support.
The authors thank for the question and have included this comment in the text: Introduction section line 148-154.
- The authors mention that the hydrogel loses weight in time; nevertheless, it is not clear whether this lost, which is noticed from day 3 (Figure 2A), would be beneficial for the organoids.
The authors clarify this point modifying the text accordingly:
“As shown in Figure 2A, the hydrogel tends to gradually lose weight over time when cultured in incubator at 37°C, 5% CO2 in neural medium. Specifically, the hydrogel's weight loss remained contained during the first weeks, while after 1 month in culture at the same conditions the hydrogel was fully degraded. The hydrogel stability during the first two weeks of iPSCs-derived cORGs generation is crucial to sustain the neural induction and to guide the internal cytoarchitecture. The proper and robust hydrogel scaffold could simulate the presence of ECM and lead neural stem cells to self-organize and differentiate into the radial structures and cortical layers. “
in
“As shown in Figure 3A, the hydrogel tends to gradually lose weight over time when cultured in an incubator at 37°C, 5% CO2 in neural medium. Specifically, the hydrogel's weight loss remained contained during the first weeks, while after 1 month in culture under the same conditions, the hydrogel was fully degraded. The hydrogel stability during the first two weeks of iPSCs-derived cORGs generation is crucial to sustain the neural induction and to guide the internal cytoarchitecture. The gradual weight loss of the hydrogel from day 3 indicates a controlled degradation process, allowing organoids to transition from a scaffold-supported state to a more self-organized structure. This mimics the in vivo extracellular matrix remodeling, facilitating nutrient diffusion, cellular expansion, and network maturation. As the hydrogel degrades, it reduces physical constraints on growing organoids while maintaining enough mechanical support early on, promoting a balance between structural guidance and autonomous tissue development. The proper and robust hydrogel scaffold could simulate the presence of ECM and lead neural stem cells to self-organize and differentiate into radial structures and cortical layers“.
- Explain in more detail why the hydrogel takes 7 days for reaching its maximum swelling (Figure 2D). A hydrogel should be completely swollen after 24 hours.
The graph reports time points below the 24 hours to show the actual trend that confirms the complete swelling time at 24hrs since the statistical intervals indicate a lack of difference between the 24hrs and the following time points. The difference is not statistically significant to be considered a further swelling after 24hrs.
The authors apologize for the misunderstanding and modify the text accordingly:
“A swelling test was used to evaluate the ability of the SA-CMC-Matrigel® hydrogel-based ink to incorporate water. As described in Figure 2D, the hydrogel is prompt to swell, up to about doubling the original weight after 24h, and absorbed liquids with a progressive increasing kinetics tending to plateau after 5 days.”
in
“A swelling test evaluated the ability of the SA-CMC-Matrigel® hydrogel-based ink to incorporate water. The hydrogel absorbs liquids with gradually rising kinetics that tend to plateau after 24 hours, and it is prone to swelling, ultimately doubling its initial weight, as seen in Figure 4B”
- The authors mention that the maturation time for the 3D printed organoid is longer compared to the manual Matrigel organoid. Please, explain why the 3D organoids were not matured for more time.
The authors thank for the observation. We set up a 40 day- protocol based on our experience and data reported in literature. The 40 days timing is considered an optimal window for cerebral organoid differentiation because it balances cellular differentiation, structural formation, and practicality in terms of cell health and reproducibility.
Prolonged differentiation could result in cellular overcrowding, altered architecture, or reduced functional development. Larger organoids often face issues with oxygen and nutrient diffusion to their core, which could result in necrosis in the center.
- Include qPCR results for different timepoints and not only after 40 days of maturation.
Unfortunately, for this Special Issue, we cannot include a qPCR analysis of time points prior to the final day of differentiation (40 days). The entire process is currently being replicated, and we think to include the data we are collecting at different time points in a new publication in this journal. The Authors thank for this suggestion but would take longer time to insert also this result.
- Include the pluripotent markers for the iPS cell line previously reprogrammed. Add this information as supplemental data.
The authors, as requested, have added supplemental data including bright field images, qPCR results of pluripotency markers, and mycoplasma test for each iPS cell lines used, and have updated the Materials and Methods section, accordingly.
- In the results and discussion part, it is recommended to include other articles and compared the obtained results with them. The manuscript lacks discussion of the results by comparing them with published literature.
The authors thank for this observation and have revised the manuscript adding discussion and other articles on published literature in this field.
11.The manuscript is well-written; nevertheless, it is recommended to check the use of abbreviations as well as the use of uppercases. In some parts, connections between ideas need to be improved.
The authors thank for this comment and have revised the entire manuscript as suggested.

Reviewer 2 Report
Comments and Suggestions for Authors
Congratulations on this work. It is well described and seems promising for future investigation. I have only a couple of suggestions to improve its presentation.
The introduction is very long and needs to be broken up into manageable paragraphs. Please review all sections for this (especially 2.1, 2.2 and 2.4).
Please add a scale bar to the bottom row of images for Figures 1C and 3C.
Author Response
Congratulations on this work. It is well described and seems promising for future investigation. I have only a couple of suggestions to improve its presentation.
The authors appreciate the Reviewer for the appreciation of our work and for agreeing to review this paper.
Comments:
The introduction is very long and needs to be broken up into manageable paragraphs. Please review all sections for this (especially 2.1, 2.2 and 2.4).
The authors thank for the observation and have revised the manuscript as suggested, shortening the introduction, and dividing sections into more paragraphs. The numbering of figures and paragraphs has been changed accordingly.
Please add a scale bar to the bottom row of images for Figures 1C and 3C.
The authors apologize for the oversight and have added the scale bar to the bottom row of images for Figures indicated.

Reviewer 3 Report
Comments and Suggestions for Authors
This study presents an innovative method for printing neural embryoid bodies using a chemically defined hydrogel composed of Matrigel®, sodium alginate, and CMC. The elimination of operator-dependent embedding and the use of a bioink containing sodium alginate and CMC for bioprinting represent a significant advancement, potentially enhancing scalability and reducing costs. The manuscript is well written and clearly explained; however, some modifications could further strengthen the rationale and provide additional clarity.
- Providing more context in the introduction regarding the specific advantages of the hydrogel composition, particularly the roles of sodium alginate and sodium carboxymethylcellulose, would strengthen the rationale for their selection.
- While the paper effectively highlights the advantages and limitations of manually Matrigel®-embedded cerebral organoids, it would benefit from a more detailed comparison with alternative embedding techniques, such as microfluidic-based approaches, which could improve reproducibility and reduce labor intensity.
- The 3D bioprinting parameters are well-documented, but it is unclear how these settings were optimized. Were preliminary tests conducted to determine optimal conditions, and how were the concentrations of NaCMC and sodium alginate chosen? A brief explanation of this would enhance the clarity of the methodology.
- The delayed neural folding in bioprinted spheroids warrants further discussion. Could differences in hydrogel mechanics or matrix composition be influencing this delay?
- Could additional factors, such as mechanical properties or cellular interactions, contribute to the observed delay in the maturation of hydrogel-embedded cORGs?
Author Response
This study presents an innovative method for printing neural embryoid bodies using a chemically defined hydrogel composed of Matrigel®, sodium alginate, and CMC. The elimination of operator-dependent embedding and the use of a bioink containing sodium alginate and CMC for bioprinting represent a significant advancement, potentially enhancing scalability and reducing costs. The manuscript is well written and clearly explained; however, some modifications could further strengthen the rationale and provide additional clarity.
The authors thank the Reviewer for the words of appreciation and for accepting to review this paper.
Comments:
- Providing more context in the introduction regarding the specific advantages of the hydrogel composition, particularly the roles of sodium alginate and sodium carboxymethylcellulose, would strengthen the rationale for their selection.
The authors thank for the observation and have revised the manuscript as suggested. Introduction section, line 148-154:
“SA and CMC were chosen to compose the hydrogels because these components offer a more controllable and biomimetic alternative to Matrigel® for brain organoid culture. SA hydrogels can be precisely adjusted in stiffness through ionic crosslinking, better matching the elasticity of human brain ECM. CMC enhances viscosity and printability, ensuring structural integrity during fabrication [34, 35]. Beyond mechanics, SA-CMC hydrogels can be potentially chemically functionalized with human-relevant ECM proteins. This customization supports stem cell adhesion, differentiation, and network formation while providing a stable and reproducible environment for long-term studies [36]”.
- While the paper effectively highlights the advantages and limitations of manually Matrigel®-embedded cerebral organoids, it would benefit from a more detailed comparison with alternative embedding techniques, such as microfluidic-based approaches, which could improve reproducibility and reduce labor intensity.
The Authors thank for this comment and have added a comparison between alternative embedding technique and manually MatrigelR-embedding cORGs in the Conclusion section, line 623-642.
“To date, a valid alternative to ECM-embedded cORGs is microfluidic-based approach. This system uses precisely engineered channels that allow for better nutrient and waste management through active fluid flow, which improves the growth and development of organoids. Such platforms can integrate more complex environmental conditions, like mechanical forces, oxygen gradients, and controlled application of signaling molecules, providing a more physiologically relevant model of brain development [53]. Automation in microfluidic systems significantly reduces labor intensity compared to manual methods, as processes like nutrient delivery and media changes can be automated, leading to more efficient and scalable experiments [54]. However, microfluidic systems are more expensive and technically challenging to set up, requiring specialized equipment and expertise. For studies that require high control, reproducibility, and scalability, especially in disease modeling or drug screening, microfluidic-based approaches may be the best option. However, for smaller-scale studies or when cost is a major factor, manually Matrigel®-embedded organoids may be sufficient, offering a simpler but still valuable platform for investigating brain development and cellular behavior. In terms of biological relevance Matrigel®-embedding being more affordable, become more widespread and this can promote comparison between different laboratories. Based on the above it is evident that the search for the perfect biological niche for iPSCs-derived brain organoid in vitro generation is still ongoing”.
- The 3D bioprinting parameters are well-documented, but it is unclear how these settings were optimized. Were preliminary tests conducted to determine optimal conditions, and how were the concentrations of NaCMC and sodium alginate chosen? A brief explanation of this would enhance the clarity of the methodology.
The concentration and printing optimization of such hydrogel derive directly from previous studies from the authors not properly mentioned in this manuscript. The authors apologize for the misunderstanding and correct the text in Material and Methods section, paragraph 4.5, line 739-750 accordingly:
“The hydrogel was developed by dissolving 1.5 w/v% SA (low viscosity, Thermo Fisher Scientific, Waltham, MA) and 1 w/v% of NaCMC (Mw= 250 kDa, Thermo Fisher Scientific) in Milli-Q water. The solution was then sterilized in an autoclave at 121°C for 20 min. When the mixture reaches a warm temperature, a proper volume (1:10 dilution) of Neurobasal Medium (Thermo Fisher Scientific, Waltham, MA) containing hiPSCs-derived nEBs at day 7 of generation and matrix Matrigel (1:10) were added. The hydrogel so composed was poured into a sterile cartridge Optimum Class VI, 30 cc (Nordson, Westlake, OH). Printing was conducted in a 12 well plate with the 3D-Bioplotter system (EnvisionTEC, DE). The experimental parameters used for the extrusion printing were pressure of 0.4 bar, speed of 100 mm/s and needle diameter of 0.6 mm. After printing, 0,5 mL of 1 w/w% CaCl2 was poured on the top of each hydrogel and kept for 20 min in an incubator at 37°C. “
in
“The hydrogel was developed by dissolving 1.5 w/v% SA (low viscosity, Thermo Fisher Scientific, Waltham, MA) and 1 w/v% of NaCMC (Mw= 250 kDa, Thermo Fisher Scientific) in Milli-Q water, as results from different hydrogel formulations demonstrated by the authors [56]. The solution was then sterilized in an autoclave at 121°C for 20 min. When the mixture reaches a warm temperature, a proper volume (1:10 dilution) of Neurobasal Medium (Thermo Fisher Scientific, Waltham, MA) containing hiPSCs-derived nEBs at day 7 of generation and matrix Matrigel (1:10) were added. Printing was conducted in a 12-well plate with the 3D-Bioplotter system (EnvisionTEC, DE). The experimental parameters used for the extrusion printing were pressure of 0.4 bar, speed of 100 mm/s and needle diameter of 0.6 mm as already described in previous works from the authors [57]. 0,5 mL of 1 w/w% CaCl2 was used for crosslinking on the top of each hydrogel and kept for 20 min in an incubator at 37°C.”
- The delayed neural folding in bioprinted spheroids warrants further discussion. Could differences in hydrogel mechanics or matrix composition be influencing this delay?
- Could additional factors, such as mechanical properties or cellular interactions, contribute to the observed delay in the maturation of hydrogel-embedded cORGs?
The authors thanks for these observations and have added clarification in the Result and Discussion section, Paragraph 2.5, line 419-432.
The mechanical properties of the hydrogel, such as stiffness, elasticity, and viscosity, play a crucial role in supporting cellular growth, differentiation, and organization. If the hydrogel is too soft or rigid, it may not provide the best mechanical cues necessary for cells to properly align, migrate, and form folds, which are essential for neural development [40]. The stiffness and viscosity of pure Matrigel provide the appropriate mechanical tension to the extracellular matrix, promoting the embedded cells to form tubular structures by arranging themselves radially around a central lumen. Conversely, the hydrogel embedding is less viscous and rigid, which can slow the generation of neural rosettes [41]. Moreover, the slowdown observed in the growth and maturation of hydrogel-based 3D printed cORGs could also be due to the lower concentration of Matrigel® present in the ink. Literature reports that Matrigel® exposure influences organoid size, morphology, and cell type composition. Particularly, it is reported that the amount of Matrigel® used in the embedding area is directly proportional to the size of the resulting brain organoid [42].

Round 2
Reviewer 1 Report
Comments and Suggestions for Authors
Thanks for answering of the observations.